# Characterization of Class-3 Semaphorin Receptors, Neuropilins and Plexins, as Therapeutic Targets in a Pan-Cancer Study

**DOI:** 10.3390/cancers12071816

**Published:** 2020-07-06

**Authors:** Xiaoli Zhang, Shuai Shao, Lang Li

**Affiliations:** 1Department of Biomedical Informatics, College of Medicine, The Ohio State University, 320B Lincoln Tower, 1800 Cannon Dr., Columbus, OH 43210, USA; lang.li@osumc.edu; 2Division of Pharmaceutics and Pharmacology, College of Pharmacy, The Ohio State University, Columbus, OH 43201, USA; shao.398@buckeyemail.osu.edu

**Keywords:** class-3 semaphorin receptors, neuropilins and plexins, gene expression, survival, tumor microenvironment, drug responses, cancer type and subtype dependent

## Abstract

Class-3 semaphorins (SEMA3s), initially characterized as axon guidance cues, have been recognized as key regulators for immune responses, angiogenesis, tumorigenesis and drug responses. The functions of SEMA3s are attributed to the activation of downstream signaling cascades mainly mediated by cell surface receptors neuropilins (NRPs) and plexins (PLXNs), yet their roles in human cancers are not completely understood. Here, we provided a detailed pan-cancer analysis of NRPs and PLXNs in their expression, and association with key signal transducers, patient survival, tumor microenvironment (TME), and drug responses. The expression of NRPs and PLXNs were dysregulated in many cancer types, and the majority of them were further dysregulated in metastatic tumors, indicating a role in metastatic progression. Importantly, the expression of these genes was frequently associated with key transducers, patient survival, TME, and drug responses; however, the direction of the association varied for the particular gene queried and the specific cancer type/subtype tested. Specifically, NRP1, NRP2, PLXNA1, PLXNA3, PLXNB3, PLXNC1, and PLXND1 were primarily associated with aggressive phenotypes, whereas the rest were more associated with favorable prognosis. These data highlighted the need to study each as a separate entity in a cancer type- and subtype-dependent manner.

## 1. Introduction

Class-3 semaphorins (SEMA3s), including seven members (SEMA3A-SEMA3G), are the only family of secreted proteins among all the ~28 identified vertebrate semaphorins. Semaphorins are characterized by the presence of a ~500 amino acids long conservative N-terminal sema domain signature which is also present in semaphorin receptors of the plexin family [1]. Similar to all other semephorins, SEMA3s have been reported to paly pivotal roles in immune responses, angiogenesis, apoptosis, cell migration, local cancer spread and metastases [2,3,4,5,6,7,8]. Most of the SEMA3s form functional class-3 semaphorin-receptor complexes to transduce their signals by binding to coreceptors of the neuropilin (NRP) family which subsequently associate with receptors of the plexin (PLXN) family [8,9,10]. In recent years, the roles of SEMA3s and their receptors NRPs and PLXNs in cancer are increasingly recognized. The SEMA3s are reported to have both pro- and anti-tumor properties that are cell type and cell context dependent [3,5,8,11,12,13]. Our previous systematic pan-cancer study confirmed the identification of SEMA3s as therapeutic targets, and demonstrated that their tumor promoter or tumor suppressor role needs to be studied in a cancer type, immune subtype, and molecular-subtype dependent manner [11]. NRPs and PLXNs are instrumental in controlling and transducing signals from SEMA3 ligands to activate downstream signaling cascades in cancer cells; however, their functions in cancer were not fully understood and there lacks a systematic analysis to understand their function in different cancer types.

NRPs are multifunctional, non-tyrosine kinase cell surface glycoproteins expressed in all vertebrates and widely conserved across species [14]. The two members of NRPs, NRP1 and NRP2, mainly function as coreceptors for SEMA3s and for members of vascular endothelial growth factor (VGEF) family of molecules by interacting with SEMA3 receptors PLXNs and VEGF receptors, respectively. NRPs play important roles in a broad range of physiological processes and pathological conditions including cancer [4,14,15,16,17,18]. NRPs show different binding affinities for different SEMA3s. For instance, while SEMA3A binds specifically to NRP1 and SEMA3F binds to NRP2-containing holoreceptors to promote tumor cell normalization and inhibit metastasis in endothelial cells [19,20], SEMA3C shows similar affinities for NRP1 and NRP2 [12]. As NRPs lack an intracellular kinase domain, they transmit signals by recruiting various transmembrane receptor kinases, including semaphorin receptor PLXNs, to form functional complexes. However, the receptor kinase is chosen by the corresponding particular extracellular ligands [8,21].

For PLXNs, which like NRPs are also type 1 transmembrane proteins, semaphorins are their primary ligands. PLXN family contains nine members that are subgrouped into four subfamilies based on their structural homology: four Type-A PLXNs (PLXA1-A4), three Type-B PLXNs (PLXNB1-B3), and single C (PLXNC1) and D (PLXND1) PLXNs [22]. To date, only a subset of these, including PLXNA subfamily, PLXNB1, PLXNB2, and PLXND1, are known to interact with SEMA3s [8,9,10,12,23,24]. However, different SEMA3s have different specificity for different PLXN receptors, and this specificity even differs under different physiological conditions and for different signal transductions [8,25,26,27]. For example, studies suggest that SEMA3A requires NRP1, PLXNA1 and PLXNA4 to form functional receptor complexes to transduce SEMA3A-induced cytoskeletal collapse, while PLXNA3 rather than PLXNA4 is required for the transduction of SEMA3A signals to trigger neuronal apoptosis [25,26]. In addition, PLXNA2 can replace PLXNA1 in SEMA3A receptors when it is highly expressed to enable plasticity in signal transduction under diverse conditions [27]. SEMA3B requires a NRP, PLXNA2 and PLXNA4 to form the receptor complex to induce cytoskeletal collapse [27]. SEMA3C, 3D, 3E, and 3G transduce signals primarily through PLXND1. SEMA3E is the only member among all SEMA3s that does not need to bind co-receptor NRPs before interacting with its receptor PLXND1 to transduce signals [28,29,30,31]. SEMA3C and SEMA3D were reported to be able to transduce signals using other or additional PLXN receptors than PLXND1 such as PLXNB1 [24,31,32]. SEMA3F was reported to require a NRP and the presence of PLXNA1 and PLXNA3 to transduce signals in endothelial cells [33,34]. Upon semaphorin binding, PLXNs will trigger the activation of PLXN-associated receptor-type or cytoplasmic non-receptor-type tyrosine kinases (TKs) to elicit divergent semaphorin-dependent signaling pathways and functional outcomes [5,35]. PLXNs have also been shown to interact with G-protein R-Ras, and Rho family GTPases such as Rnd1 and Rac to transduce signals [35,36,37,38]. These tyrosine kinases and GTPases are considered as key transducers in triggering semaphorin downstream signaling cascades in a cell-specific manner [5,35].

In this study, we used a similar systematic pan-cancer analyses method to our study of SEMA3s to elucidate the function of NRPs and PLXNs in 33 different cancer types [11]. Our results showed that the expression of NRPs and PLXNs were also dysregulated in cancer tumors and the dysregulation was more dramatic in metastatic tumors. The expression of SEMA3 receptors showed both positive and negative association with key signal transducers, reflecting their function in a cell-specific manner. Furthermore, the expressions of NRPs and PLXNs were associated with immune subtypes, level of immune cell and stromal cell infiltrates in the tumor microenvironment (TME), and they were frequently associated with patient survival and drug responses. By studying these genes in a breast cancer cohort, we further revealed that the function of each individual receptor differs not only in a cancer type, but also in a molecular subtype- and immune subtype-dependent manner. In conclusion, our study confirmed the roles of NRPs and PLXNs as potential therapeutic targets in cancer treatment and highlighted the need to study each as a separate entity in different cancer types.

## 2. Results

### 2.1. Neuropilins and Plexins Are Dysregulated in Primary Tumors and with Further Dysregulation in Metatstatic Tumors

In our previous pan-cancer study of SEMA3s, we found they are dysregulated in multiple different cancer types, which is a defining feature of genes to function in tumorigenesis [11]. Here we investigated the expression levels of NRPs and PLXNs in all 33 available cancer types in TCGA pan-cancer data (Summery of TCGA data are in Appendix A). We also included the analysis of SEMA3s in this study to investigate whether their expression is comparable to their receptors. Similar to SEMA3s, the expression levels of the SEMA3 receptors also showed great inter- and intra-tumor heterogeneity (Figure 1A and Appendix A). Both NRPs, PLXNA1-A3, PLXNB1-B2, and PLXND1 had relatively higher expression and smaller heterogeneity, while genes PLXNA4, PLXNB3 and PLXNC1 had lower expression across all cancer types but with higher heterogeneity. NRP1, PLXNA2, PLXNA4, and SEMA3B-3E and SEMA3G showed a significant decrease (*p* < 0.0001), PLXNA1, PLXNA3, and PLXNB3 showed a significant increase (*p* < 0.0001), while NRP2, PLXNB1, PLXNB2, and PLXNC1, SEMA3A and SEMA3F showed no overall significant difference in tumors compared to that of normal samples averaged across all 33 cancer types (*p* > 0.05) (Figure 1A). Notably, the majority of the genes that showed differential expression in tumors, except PLXNA3, SEMA3B and SEMA3D, were further dysregulated in the metastatic tumors, where PLXNA1 and PLXNB3 were further upregulated, whereas genes NRP1, PLXNA2, PLXNA4, SEMA3E, and SEMA3G were further downregulated in metastatic tumors (*p* < 0.0001). Interestingly, even the genes that showed no differential expression in tumors compared to normal samples also showed significantly increased (NRP2 and PLXNC1) or decreased (PLXNB1 and SEMA3F) expression in metastatic tumors (*p* < 0.0001). These results suggest that these genes play crucial roles not only during tumorigenesis but also during tumor metastasis (Figure 1A).

In order to understand the dysregulation of NRP and PLXN genes in individual cancer types, next we compared the gene expression in the tumor of each cancer type to their corresponding normal samples using the 17 cancer types that had at least five paired adjacent normal samples (Figure 1B). The heatmap showed that the expression of SEMA3s and their receptors were separated into two major clusters. One cluster includes SEMA3D, SEMA3E, SEMA3G and PLXNA4, which all showed primarily down-regulation in the tested cancer tumors. The other major cluster is further separated into several subclusters, with one subcluster including PLXNA1, PLXNA3, and SEMA3F showing predominantly increased expression in cancer tumors. The rest of SEMA3s and receptors showed mixed up- and down-regulation in different cancer types. Apparently, the heatmap also shows that the expression of SEMA3s is not necessary to have a similar pattern to their receptors. For example, PLXND1 was reported to be the receptor for SEMA3C-SEMA3E, but their expression was not significantly correlated across cancer types (*p* > 0.05) and they were not clustered together (Figure 1B).

Because protein is the final product of gene expression that links genotypes to phenotypes, we further tested whether expression of SEMA3s and their receptors are significantly correlated at the corresponding mRNA and protein level. Three TCGA cancer types that have both RNA-Seq and proteomics data, i.e., breast cancer (BRCA) [39], ovarian cancer (OV) [40], and colorectal cancer (COAD) [41] were used for the test. By utilizing genes that had both mRNA and proteomics data, Pearson Correlation test showed that gene expression is not always agreed at the mRNA level and protein level in the three cancer types tested (Figure 1C and Appendix A). Although there is a general positive correlation pattern, the degree of correlation varies for different genes in different cancer types. A higher correlation in BRCA and OV than in COAD was observed, which could be due to the relatively low quality and lower number of genes detected in COAD proteomics study [41].

### 2.2. Expression of Neuropilins and Plexins Are Associated with Semaphroin Key Transducers

PLXNs need to activate PLXN-associated tyrosine kinases (TKs) or interact with intracellular signal transducers R-Ras or Rho family GTPases to transduce semaphorin-dependent signaling pathways [5,35]. Therefore, we tested whether the expression of SEMA3 receptors are significantly associated with the expression of these TKs and GTPases. We included receptor-type TKs (MET, ERBB2, RON (MST1R), VEGFR2 (KDR), and OTK (PTK7)), and cytoplasmic non-receptor type TKs (FYN, FES, PYK2, FAK (PTK2) and SRC), as well as R-RAS, RhoA, Rac1 and Rnd1 in the analysis. Associations between these genes and SEMA3 receptors showed high variation in a cancer type- and specific gene-dependent manner (Figure 2 and Appendix A). Two major clusters were observed from the heatmap generated based on the correlation coefficient between SEMA3 receptors and these key transducers averaged across all 33 tested cancer types (Figure 2). In general, PLXNA4, PLXNC1 and PLXND1 showed positive association with FES, VEGFR2, FYN, RhoA, PYK2, and Rnd1, while negative association with ERBB2, MET, R-RAS, FAK, Rac1, OTK, Ron, and SRC. The rest of the SEMA3 receptors showed opposite correlations with these semaphorin key signal transducers (Figure 2). Specifically, PLXND1 showed significant positive association with FES (r = 0.47), PLXNA1 with SRC (r = 0.46) and OTK (r = 0.38), NRP1 with VEGFR2 (r = 0.67) and MET (r = 0.36), NRP2 with VEGFR2 (r = 0.46) and FYN (r = 0.46), PLXNB2 with ERBB2 (r = 0.41) and RRAS (r = 0.36), PLXNB1 with ERBB2 (r = 0.35), and PLXNA2 with VEGFR2 (r = 0.38).

### 2.3. Expression of Neuropilins and Plexins Are Associated with Patient Survival

Our previous study showed that SEMA3s were generally correlated with patient overall survival [11]. Because the functions of SEMA3s are attributed to their receptors, we further tested whether the expression of SEMA3 receptors could also predict patient survival outcomes, including overall survival (OS) and progression free interval (PFI). For the survival analysis, all 33 cancer types were tested with univariate Cox proportional hazard regression models, and we claimed significant association with a *p*-value < 0.05 without adjustment for multiple comparisons to be consistent with the display of the results with forest plots (Figure 3 and Appendix A). The results showed that each of the NRPs and PLXNs was significantly associated with the survival of a number of cancer types; however, the direction of the association varied depending on the member queried and the cancer type tested. A general pattern can be observed that increased gene expression is normally associated with worse survival, and vice versa. For example, PLXNA1 was upregulated in majority of the tested cancer tumors, and the increased expression was mainly associated with increased disadvantage for both OS and PFI (Figure 3 and Appendix A). It predicted OS disadvantage for patients with adrenocortical cancer (ACC), kidney clear cell carcinoma (KIRC), acute myeloid leukemia (LAML), liver hepatocellular carcinoma (LIHC), mesothelioma (MESO), ovarian cancer (OV), sarcoma (SARC), and uterine corpus endometrioid carcinoma (UCEC). It also associated with worse PFI for ACC, colon adenocarcinoma (COAD), KIRC, MESO, pancreatic adenocarcinoma (PAAD), and prostate adenocarcinoma (PRAD). PLXNA3 was another gene upregulated in the majority of the tested cancers and the increased expression was associated with worse OS and PFI of COAD, KIRC, brain lower grade glioma (LGG), MESO, SARC, UCEC, uveal carcinosarcoma (UVM), and PRAD. PLXND1 showed mixed up- and down-regulations in different cancer types, and its expression favored OS advantage for lung adenocarcinoma (LUAD), thymoma (THYM), and UVM, and survival disadvantage for breast carcinoma (BRCA), COAD, kidney papillary cell carcinoma (KIRP), LGG, lung squamous cell carcinoma (LUSC) and MESO. Although PLXNA4 was predominantly downregulated in cancer tumors, it was not associated with the survival of majority of the cancer types. For the rest of the PLXN family members, PLXNA2 was only associated with survival advantage for KIRC and LUAD, and PLXNC1 was only associated with survival disadvantage of LAML and stomach adenocarcinoma (STAD). The PLXNB subgroup were associated with both survival advantage and disadvantage of a number of cancer types, but PLXNB1 and PLXNB2 were more associated with better survival, while PLXNB3 was more associated with worse survival (Figure 3 and Appendix A). Increased expression of NRP1 and NRP2 was mainly associated with poor prognosis: NRP1 associated with poor OS for ACC, cervical & endocervical cancer (CESC), LGG, MESO, and STAD, while NRP2 associated with poor prognosis for bladder urothelial carcinoma (BLCA), KIRP, MESO, PAAD, and STAD, respectively. However, the results showed that the direction of the dysregulated gene expression was not always consistent with the direction of their association with survival outcomes. For instance, PLXNB1 and PLXNB2 had increased expression in UCEC but they were associated with favorable OS, instead, PLXND1 had decreased expression in UCEC but it was associated with poor prognosis for OS (Figure 1B and Figure 3). Another example is NRP2, it had decreased expression in BLCA and STAD, but increased expression of NRP2 was associated with survival disadvantage for patients with BLCA and STAD. These contradictory results need to be further investigated.

### 2.4. Neuropilins and Plexins Are Associated with Immune Infiltration Subtypes and Tumor Microenvironment in Cancer

Similar to SEMA3s, where SEMA3A and SEMA3E are among the so-called “immuno” semaphorins [42,43,44,45], SEMA3 receptors NRPs and PLXNs also play important roles in immune responses [14,46]. For example, both NRPs have shown to be expressed in multiple types of immune cells, such as dendritic cells (DCs), macrophages, T cell subpopulations, and crucial to regulate immune responses under normal as well as clinical conditions [14]. PLXNA1 can interact with SEMA3A to regulate DC cell movement [47], and interact with SEMA6D to play important roles in immune response and bone homeostasis [48]. In addition, PLXNB1 was found to be important in controlling germinal center formation and long-term B cell immune responses [49]. In this study, we extended our previous study of SEMA3s [11] to SEMA3 receptors to test the correlation between their expression and tumor infiltrates, as well as their association with the expression of immune blockade molecules PD1/PDL1 and CTLA-4 in cancer TME. PD1/PDL1 and CTLA-4 have been used as immunotherapeutic targets in a number of cancer types with high expression of these genes.

By using the Cancer Genome Atlas (TCGA) pan-cancer data, six types of immune infiltrates were identified across all cancer tumors that correspond from tumor promoting to tumor suppressive [50]. They are C1 (wound healing), C2 (INF-r dominant), C3 (inflammatory), C4 (lymphocyte depleted), C5 (immunologically quiet), and C6 (TGFβ-dominant). Among the six immune infiltrate subtypes, patients characterized into C3 and C5 immune subtypes had significantly better survival than those belonging to other immune subtypes, where patients belonging to C4 and C6 subtypes had the least favorable survival [11,50]. We previously showed that SEMA3A, SEMA3C, SEMA3E and SEMA3F were associated with more aggressive immune subtypes (C1, C2, and C6), while the other SEMA3s were associated with immune subtypes predicting better survival [11]. Here, we compared the expression of NRPs and PLXNs among different immune infiltrate subtypes. Different NRP and PLXN genes showed distinct association patterns with different immune subtypes (Figure 4A). NRP1, PLXNB2, and PLXND1 showed increased expression in C3 and C6, and decreased expression in C5, while PLXNB3 had increased expression in C4 and C5, indicating they may play both tumor promoter and tumor suppressor roles as C3 and C5 subtypes associated the most with better survival, but subtypes C4 and C6 associated with worse survival, respectively. PLXNA4 and PLXNB1 had significantly increased expression in subtype C5, suggesting an association between higher gene expression and favorable immune infiltrate type, indicating a tumor suppressor role. In contrast, PLXNA1 showed higher expression in C1, C2 and C6, NRP2 and PLXNC1 had increased expression in C6, indicating that these genes may mainly play a tumor promoter role as patients characterized into those subtypes had worse survival due to higher proliferation rate and enrichment of TGFβ [11,50]. PLXNA2 and PLXNA3 showed no significant differential expression among all six subtypes (Figure 4A). In summary, the distinct expression patterns of SEMA3 receptors in different immune subtypes suggest that the function of each gene is also immune subtype dependent.

As secreted proteins, SEMA3s can be released from tumor cells as well as from cells in the TME, such as infiltrating leukocytes, to regulate tumor growth and metastasis by interacting with their receptors through autocrine regulatory loops [5,14]. Therefore, we further investigated the association between the expression levels of SEMA3 receptors with the levels of stromal cell and immune cell infiltrations in the TME, and tumor purity in tumors indicated by stromal scores, immune scores, and estimate scores using the algorithm, ESTIMATE (Figure 4B and Appendix A) [11,51,52]. The majority of the tested cancer types showed positive correlation between the expression of NRP1, NRP2, PLXNC1 and PLXND1, negative correlation between expression of PLXNB1, and the three scores. For stromal score, NRP1 showed the highest positive correlation across all cancer types (*r* = 0.59), followed by PLXND1 (*r* = 0.54), NRP2 (*r* = 0.50), PLXNC1 (*r* = 0.48) and PLXNA4 (*r* = 0.28) (*p* < 0.0001). For immune score, PLXND1 showed the highest correlation (*r* = 0.46) across all cancer types, followed by PLXNC1 (*r* = 0.39), NRP1 (*r* = 0.23) and NRP2 (*r* = 0.15) (*p* < 0.0001). The order of the correlation between the expressions of NRPs and PLXNs and estimate score is the same as that for stromal score, but with different correlation coefficients (Figure 4B and Appendix A). For all three scores, PLXNB1 showed the strongest negative correlation, followed by PLXNB3 and PLXNA3. The other receptors showed very trivial correlation with the three scores, even though all the *p*-values were significant due to the large sample size of the TCGA pan-cancer data. These results indicate that different cancer types had different levels of presence of these genes in stromal cells and/or immune infiltrating cells in the TME. High immune infiltration in TME is normally correlated with poor prognosis, but high immune T-cell infiltration, provides opportunity for patients to be treated with immune checkpoint inhibitors such as anti-PD1/PDL1 and anti-CTLA4 inhibitors [53]. Therefore, we further tested the correlation between the expression of NRPs and PLXNs and immune checkpoint molecules PD1, PDL1, and CTLA-4. The results showed that NRP1, NRP2, PLXNC1, and PLXND1 also positively correlated, while PLXNB1 negatively correlated with levels of PD1, PDL1, and CTLA-4 gene expression (*p* < 0.0001) (Figure 4C and Appendix A). Therefore, NRPs and PLXNs especially the receptors showing high positive correlation with TME and immune checkpoint molecules may be potential targets and/or immune modulators for cancer treatment.

### 2.5. Neuropilins and Plexins Are Associated with Tumor Stemness and Cancer Cell Response to Chemotherapy

Cancer stem cells (CSCs) have been considered responsible for tumor recurrence, metastasis, and chemotherapy resistance [54]. A couple of SEMA3 members such as SEMA3A and SEMA3C and members of NRPs and PLXNs have been found to have high expression in CSCs or CSC-like cells, and to be involved in drug resistance and poor prognosis in multiple cancer types [55,56]. For example, it was reported that the functional complex of SEMA3C with its receptors NRP1/PLXNA2/PLXND1 could activate Rac1/NF-κB signaling to promote the survival and migration of glioma stem-like cells [57]. The measurement of tumor stemness was developed using TCGA pan-cancer data based on mRNA expression (RNA stemness score: RNAss) and DNA methylation pattern (DNA stemness score: DNAss) [58]. Here, we explored the correlation between the expression of NRPs and PLXNs with tumor stemness scores. Similar to SEMA3s, members of NRPs and PLXNs also showed a broad range of levels of association with RNAss and DNAss in different cancer types (Figure 5A,B) [11]. All SEMA3 receptors, except PLXNA1 which showed negligible positive correlation, were negatively correlated with RNAss score with varying degrees (correlation coefficients *r*: −0.65 to −0.13) (*p* < 0.0001), where NRPs showed the strongest association (*r* = −0.65 and −0.48 for NRP1 and NRP2, respectively) across cancer types. For the correlation with DNAss score, PLXNA1 and A2 showed positive correlation (*r* = 0.38 and 0.18, respectively), while all other members showed negligible (|r| < 0.17) or insignificant correlation (PLXNA3 and NRP2) across all cancer types. At the individual cancer type level, we found that NRP2 and PLXNA2 were positively, while PLXNB1 and C1 were negatively correlated with ovarian cancer stemness score measured with DNAss, but not with RNAss. In pheochromocytoma & paraganglioma (PCPG), PLXND1 was negatively, while PLXNB1 and B3 were positively correlated with both RNAss and DNAss. In addition, similar to what we observed for SEMA3s [11], all genes showed different degrees of negative correlation with both DNAss and RNAss for TGCT, however, genes showed opposite correlation with DNAss and RNAss in THYM. These contradictory results suggest that RNAss and DNAss may identify distinct cancerous cell populations characterized by different features or degrees of stemness in different cancers [58].

SEMA3s and their receptors have been reported to be related with drug responses in a number of cancer types [3,59,60,61]. We next investigated the expression of NRPs and PLXNs in NCI-60 cell lines and systematically tested the correlation between their expression levels with drug sensitivity score of over 200 chemotherapy drugs. We observed that levels of these genes showed great heterogeneity in different cell lines as in patient tumors (Appendix A), with PLXNA1 having the highest expression and PLXNA4 having the lowest expression across all cancer cell lines. For the drug response in the NCI-60 cell line data, drug sensitivity was measured by *Z*-scores, and the higher the score, the more sensitive the cells are to the drug treatment [62]. Positive correlation between increased gene expression and drug sensitivity score means increased expression corresponds to good drug response, and vice versa. Based on the correlation between gene expression and drug response *Z*-scores, we found that members of NRPs and PLXNs showed strong correlation with drug responses (|r| ≥0.4 and *p* < 0.0001), especially NRP1, NRP2, PLXNA1, PLXNA2, PLXNB1, PLXNB2, PLXNB3, and PLXNC1, although the direction of the correlation varies for different genes with different drugs (Figure 5C and Appendix A). A few genes such as NRP1, NRP2, and PLXNA1, PLXNB1 showed both positive correlation with a number of drugs and negative correlation with some other drugs. We also found that PLXNA2 and PLXNB2 mainly correlated with increased resistance, while PLXNB3 and PLXNC1 mainly correlated with increased drug sensitivity to a list of drugs. Interestingly, we noticed that NRP1, NRP2, PLXNA1, PLXNB3, and PLXNC1 were commonly correlated with several drugs. For instance, both PLXNA1 and PLXNC1 showed positive association with increased sensitivity of cells to Bafetinib, while NRP1 showed association with increased cell resistance to the same drug. In addition, NRP2, PLXNA1, PLXNC1, and PLXNB3 were found to be all positively associated with cell sensitivity to Vemurafenib, which is used for the treatment for late-stage melanoma. Furthermore, NRP2, PLXNC1 and PLXNB3 were also positively associated with cell sensitivity to Dabrafenib, which is the treatment for late-stage melanoma and metastatic non-small cell lung cancer with BRAF V600E or V600K mutations (Figure 5C and Appendix A). Therefore, Bafetinib, Dabrafenib and Vemurafenib might be good treatment option for cancer patients who have high expression of these aforementioned genes.

### 2.6. Neuropilins and Plexins in Breast Cancer

#### 2.6.1. Neuropilins and Plexins Are Dysregulated in Breast Cancer and Their Expression Is Tumor Subtype Specific

We previously provided thorough investigation of SEMA3 genes in the TCGA breast cancer patient cohort due to the fact that most of SEMA3 members have been studied or at least partially studied in breast cancer, mainly based on cell line and animal model studies [11,12,63,64,65,66,67,68,69,70,71,72,73,74]. Here we investigated how neuroplilins and plexins work coordinately with SEMA3s to promote or inhibit tumorigenesis in breast cancer tumors. Our analysis showed that PLXNA3, B2, and C1 were significantly upregulated, and NRP1, NRP2, PLXNA1, A2, A4 were significantly downregulated, while the rest of the members (PLXNB1, B3 and D1) were not differentially expressed in breast cancer tumors (Figure 6A). In addition, PLXNA3, B2, and B3 had further increased, while NRP2 and PLXNA2 had further decreased expression in metastatic tumors than in primary tumors although none of the differences were significant due to the small number of metastatic tumor samples (*n* = 7) (Figure 6A). We further compared the expression of NRPs and PLXNs among the five breast cancer molecular subtypes, i.e., basal-like (Basal), Her2 enriched (Her2), luminal A (LumA), luminal B (LumB), and Normal like (Normal), where patients characterized into Basal and Her2 subtype have the least favorable survival. Interestingly, all the genes with decreased expression in breast cancer tumor, i.e., NRP1, NRP2, PLXNA1, A2, and A4, showed higher expression in normal-like tumors than in other more aggressive subtypes, although PLXNA1 and A2 also showed higher expression in basal type tumors, one of the most aggressive type of breast cancer subtypes (Figure 6B). In contrast, PLXNB3 showed higher expression in both Basal and Her2 subtypes, indicating PLXNB3 may associate with more aggressive tumors as it also showed higher expression in metastatic than in primary tumors (Figure 6A,B). For the rest of the receptors, PLXNB1 had higher expression in LumA, PLXNB2 had higher expression in Her2, and both PLXNC1 and D1 had significantly decreased expression in Basal type but increased expression in Her2 subtype. We also observed the same phenomenon in other cancer types, such as in BLCA, head and neck squamous cell sarcoma (HNSC), KIRC and OV, that the expression of each receptors varies in different molecular subtypes within the same tumor origin (Appendix A). In summary, these results indicate that the role of each specific SEMA3 receptor is cancer subtype dependent.

#### 2.6.2. Neuropilins and Plexins Are Associated with Tumor Microenvironment and Patient Survival in Breast Cancer

Various cell types in the TME of breast cancer tumors might contribute to the dysregulated expression of SEMA3 and their receptors. By studying different immune subtypes in breast cancer, we found that the expression of NRPs and PLXNs in different immune subtypes of breast cancer samples showed similar expression pattern to what was observed across all 33 TCGA cancer types, except that no samples were characterized into immune subtype C5 in the breast cancer cohort (Figure 4A and Figure 7A). PLXNA2, A3, B2, and B3 showed either no differential expression or a small scale of differential expression in different immune subtypes. The rest of the receptors showed significantly increased expression in either C3 or C6 immune subtypes or in both of them, and some of them also showed decreased expression in C4 (*p* < 0.0001). Except PLXNA3, B1, B2, and B3 which was either not associated or negligibly associated with the three scores in TME, i.e., immune score, stromal score, and tumor purity score, other receptors were all positively correlated with the three scores (*p* < 0.0001) in breast cancer. NRP1, NRP2, PLXNC1 and D1 showed the strongest correlation (*r* > 0.35). These results were also similar to what was observed by using all the TCGA pan-cancer tumor data (Figure 4B, Figure 7B and Appendix A, Appendix A). Lastly, we found that NRP1, NRP2, PLXNA1, C1, and D1 had the highest correlation with the expression of immune checkpoint molecules PD1/PDL1 and CTLA-4 in breast cancer. (Appendix A). These results indicate that majority of the NRPs and PLXNs may have been expressed by tissue stroma or immune cells in breast cancer, and they might be good targets or to aid for immunotherapy.

Even though NRP1, NRP2, PLXNA1, A2, A4 had decreased expression in breast cancer tumors, their deregulation was not significantly associated with OS or PFI. Increased expression of these genes (except PLXNA4) was associated with worse prognosis for PFI, which could be due to their higher expression either in aggressive molecular subtypes (PLXNA1 and PLXNA2) or immune infiltrate subtypes (NRP1, NRP2, and PLXNA1) or high association with tumor infiltrates (NRP1, NRP2, and PLXNA1) (Figure 3, Appendix A and Figure 7A–C). Increased expression of PLXNA3 and B2 associated with worse OS and PFI (not significant). Surprisingly, genes PLXNB1, B3, and D1 showed no significant differential expression in breast cancer tumors compared to normal, but PLXNB1 showed favorable prognosis, and PLXNB3 and PLXND1 showed poor prognosis for both OS and PFI. This might be explained by the fact that PLXNB1 had increased expression in less aggressive molecular subtypes (LumA and Normal subtype) and immune infiltrate subtype (C3), while PLXNB3 and PLXND1 had increased expression in more aggressive breast cancer subtypes (Basal and Her2), immune infiltrate subtypes (C1, C2, or C6), and/or in metastatic tumors (PLXNB3). Except PLXNA1, A3, and B3, other SEMA3 receptors were all negatively associated with RNA stemness score (*r* = −0.24 to −0.55, and *p* < 0.0001) (Figure 7B and Appendix A). In contrast, all of the receptors were not or slightly associated with DNAss (absolute correlation coefficient *r* < 0.12) (Figure 7B). These results showed that the aggressiveness of the genes was not necessary associated with the tumor stemness, but they do show a molecular and immune subtype dependent manner.

## 3. Discussion

The semaphorin/neuropilin/plexin complexes control a wide range of biological processes and deregulation of these complexes is associated with multiple pathological statuses. In recent years, their roles in tumor growth, metastatic spread, and drug resistance are increasingly recognized. They have been found to be involved in controlling tumor cell viability, apoptosis, proliferation, adhesion, migration, and invasion by influencing both the tumor compartment and their microenvironment. Our previous study on the comprehensive characterization of SEMA3s showed that SEMA3A, SEMA3C, SEMA3E, and SEMA3F are more likely to promote tumorigenesis and associate with poor prognosis, while the other SEMA3s are more likely to play a tumor suppressor role and generally associate with better prognosis [11]. Due to the pivotal roles of NRPs and PLXNs in transducing SEMA3 signals to activate downstream signaling cascade, in this study, we extended the systemic analysis to the primary SEMA3 receptors, NRPs and PLXNs.

By including the gene expression analysis of SEMA3s with their receptors in this study, we noticed that the expression of the ligand is not necessarily positively correlated with that of their receptors. For example, PLXND1 is the receptor for SEMA3C, 3D, and 3E [28,29,30,31], but the expression of PLXND1 is not significantly correlated with its ligands. How the level of individual SEMA3 ligand matches to that of corresponding NRP and PLXN receptors when they form functional complexes could be due to post-transcriptional and/or post-translational modification in a cell context-dependent manner. This was also demonstrated by our results that the expression of SEMA3s and their receptors at mRNA level are not always significantly correlated with their protein level as shown in BRAC, COAD, and OV. In addition, we also found that the expression of SEMA3 receptors were not always significantly associated with key signal transducers tyrosine kinases and GTPases that are needed by PLXNs to elicit semaphorin-dependent downstream signaling and functional outcomes.

Similar to SEMA3s, the expression of the receptors not only showed great heterogeneity among different tumor types, but also showed high heterogeneity within each histological subtype/tissue of origin, as well as in different immune subtypes. Our results demonstrated that the majority of the receptors showed great expression variation in different molecular subtypes of BRCA, HNSC, BLCA, OV and KIRC. At the individual gene and cancer type level, PLXNA1 and PLXNA3 were found to be predominantly up-regulated, PLXNA4 was predominantly down-regulated, while NRPs and the rest of the PLXNs showed mixed up- and down-regulation in the 17 tested cancer types, although overall NRP1 expression was downregulated, and PLXNB3 was upregulated averaged across all the 33 cancer types. Strikingly, almost all the NRP and PLXN members had more dramatic expression in metastatic tumors than in primary tumors in the pan-cancer data, indicating they play important roles not only during tumorigenesis but also during tumor metastatic progression. In fact, SEMA3E and its receptor PLXND1 was reported to promote tumor invasiveness and metastatic spreading of melanoma in mice, and SEMA3A was found to suppress tumor growth and metastasis in melanoma [75,76]. SEMA3D and PLXND1 was found to promote perineural invasion and metastasis of orthotopic pancreatic tumors in mice [77]. In addition, NRP1, SEMA3A and SEMA3F was reported to correlate with hematogenous metastasis in salivary adenoid cystic carcinoma [78].

The dysregulated expression of NRPs and PLXNs in cancer tumors was generally associated with patient overall survival and progression free interval in 33 cancer types, while the direction of association is dependent on the cancer type tested and the genes queried. In general, the upregulated gene expression was associated with worse survival, while the downregulated expression was associated with better prognosis. For example, PLXNA1 and PLXNA3, which were mainly upregulated in tumors, were mainly associated with poor prognosis, while NRPs, PLXNB3, PLXNC1 and PLXND1 had mixed association with survival (both advantage and disadvantage) that is cancer type specific, however, majority of the significant survival association were toward poor prognosis. The rest of the PLXNs, i.e., PLXNA2, A4, B1 and B2 were more associated with better survival. However, in some circumstances, we observed that decreased gene expression in cancer tumors associated with increased survival risk and vice versa which could be explained by the fact that the expression of NRPs and PLXNs were also molecular subtype- and immune infiltrate subtype-dependent within the same tumor origin. Regardless of the direction of the association, notably, almost all the members of NRPs and PLXNs were associated with both OS and PFI in UVM, which is worth further investigation.

SEMA3s and their receptors NRPs and PLXNs are a new class of immunoregulatory molecules with distinct functions in various phases of the immune responses, which have been reported to play important roles in cancer-associated immune responses [14,43]. By studying the expression of NRPs and PLXNs in different immune infiltrate subtypes in TME, we found that both NRPs, PLXNA1, PLXNB2, PLXNC1, and PLXND1 were correlated with more aggressive subtypes of immune infiltrates, i.e., C1, C2, and C6, indicating a correlation with poor prognosis. In addition, NRP1, PLXNB2, and PLXND1 also showed increased expression in immune subtype C3, which is correlated with better prognosis, indicating these genes may also play tumor suppressor roles in certain conditions. PLXNA4, PLXNB1, PLXNB3 showed highest expression in immune infiltrate subtype C5, which is associated with better prognosis. PLXNA2 and PLXNA3 showed no differential expression in different immune infiltrate subtypes. In addition, NRPs and PLXNs also correlated with the level of stromal cell infiltrates and immune cell infiltrates with various degrees based on the ESTIMATE algorithm, where NRP1, NRP2, PLXNC1 and PLXND1 had the highest correlation with the level of both stromal and immune cell infiltrates. Normally positive correlation with immune or stromal infiltration associates with worse prognosis, which is consistent with what we observed that NRP1, NRP2, PLXNC1 and PLXND1 were mainly associated with worse survival. Those findings are supported by the results of previous reports that SEMA3s, by working with their receptors, can function as pro-inflammatory and immune modulators [8,12,13,43], and they may be used as direct therapeutic targets or help predict the efficacy of immune checkpoint modulators in cancer patients. Indeed, we found that the levels of NRP1, NRP2, PLXNC1 and PLXND1 were significantly correlated with the levels of PD1/PDL1, and CTLA-4, the immune checkpoint molecules, which further confirmed their potential as immune checkpoint modulators.

Cancer stem-like-cells (CSC) promote cancer progression due to the capacity for self-renewal and invasion, and it is the main cause of treatment-induced drug resistance [79,80,81]. However, in the present study, the association between NRPs and PLXNs and stemness scores showed that the function of these genes as tumor promoter or tumor suppressor is not necessarily associated with the tumor stemness. Based on our previous results on their expression, association with patient survival and tumor microenvironment, both NRPs were mainly associated with tumor promoter roles, but they showed negative or insignificant correlation with tumor stemness scores measured by RNAss and DNAss. Therefore, the stemness score as a measurement of tumor stemness needs to be further assessed and the function of members of NRPs and PLXNs in tumor stemness involvement needs to be further investigated with different methods. Regardless of the association of gene expression with tumor stemness scores, members of NRPs and PLXNS did show strong correlation with drug responses, with NRP1, NRP2, PLXNA1, PLXNA2, PLXNB1, PLXNB2, PLXNB3 and PLXNC1 showing the highest frequency of the correlation with cell sensitivity or resistance to a number of chemotherapeutic drugs. Increased expression of NRP2, PLXNA1, PLXNB3, and PLXNC1 were all associated with increased cell sensitivity to Vemurafenib, and increased expression of NRP2, PLXNB3, and PLXNA1 were associated with sensitivity to Darafenib, which provides a great treatment option for patients with high expression of these genes. Importantly, these genes were upregulated overall in the tumor samples and metastatic samples.

By studying NRPs and PLXNs in breast cancer in detail, we observed that expressions of these genes vary in different molecular subtypes and immune subtypes within the same tumor origin and confirmed that the tumor suppressor or promoter role could be subtype specific. For instance, even though genes PLXNB1, PLXNB3, and PLXND1 showed no significant differential expression in breast cancer tumors compared to normal, their expression was associated either with less aggressive breast cancer subtypes (PLXNB1) or with more aggressive subtypes or had further deregulated expression in metastatic tumors (PLXNB3 and PLXND1). This is corresponding to their prediction of patient survival outcomes with PLXNB1 showing favorable prognosis, while PLXNB3 and PLXND1 show poor prognosis for both OS and PFI in breast cancer tumors.

In summary, based on our systematic analyses of NRPs and PLXNs in terms of their expression, association with patient survival, immune subtype, and tumor infiltration, we found that NRP1, NRP2, PLXNA1, PLXNA3, PLXNB3, PLXNC1, PLXND1 were mainly associated with poor prognosis. All these genes, except NRP1, had overall upregulated expression in primary tumors and further upregulated expression in metastatic tumors, were all associated with at least two of the following features: poor survival, more aggressive immune subtypes, and higher tumor infiltration. Interestingly, among these genes, only NRP1 showed significantly decreased overall expression in cancer tumors and further decreased expression in metastatic tumors. However, NRP1 had the highest expression in one of the most aggressive immune subtypes (C6), and was among the genes that showed the strongest association with level of immune infiltration and stromal cells, as well as significantly associated with immune checkpoint molecules. These results indicate that NRP1 may have increased level in TME and explains why it is associated with poor prognosis. In fact, previous studies showed that the upregulation of NRPs often correlate with poor patient prognosis [82,83,84,85,86]. PLXNA1 and PLXNA3 were also reported to promote tumor growth and associate with poor prognosis [57,87,88,89,90,91]. The most reported was PLXND1 in terms of the association with poor prognosis [12,22,23,73,92,93]. It is also not surprising that NRP1, NRP2, PLXNA1, PLXNA3, and PLXND1 were associated with poor prognosis, as their primary ligands SEMA3A, SEMA3C, SEMA3E, and SEMA3F were found to be predominantly functioning as tumor promoters based on previous published literatures and our previous pan-cancer analysis of SEMA3s [8,11,12,43]. However, on the other hand, the increased expression of these genes may benefit patients from treatment with Bafetinib, Dabrafenib, Hypothemycin, and Vemurafenib as these genes mainly showed positive correlation with increased sensitivity to these drugs. The rest of the PLXNs, i.e., PLXNA2, A4, B1, and B2 were overall either down-regulated or insignificantly differentially expressed in cancer tumors, but all had more decreased expression in metastatic tumors than in primary tumors. They had mixed association with survival outcomes, but tended more toward better prognosis, and they were expressed higher in immune infiltrate subtypes that were associated with better survival. Taken together, PLXNA2, A4, B1, and B2 may mainly associate with tumor suppressor roles in different cancer types and predict better survival. Surprisingly though, increased expression of PLXNA2 and PLXNB2 were primarily associated with increased cell resistance to a number of drug treatments, which seems to be contradictory to their primary roles as tumor suppressors. However, the tumor promoter or tumor suppressor roles of each individual members of NRPs and PLXNs are all cell-context dependent and need to be studied individually in a cancer type- and subtype-dependent manner.

## 4. Methods

### 4.1. Datasets Used

TCGA pan-cancer data, including RNA-Seq (RNA SeqV2 RSEM), clinical data, stemness scores based on mRNA (RNAss) and DNA-methylation (DNAss), and immune subtypes were downloaded from xena browser (https://xenabrowser.net/datapages/). The RNA-Seq data was the version that was last updated on 29 December 2016. The tumor samples in TCGA were surgical resection samples obtained from primary tumors that received no prior neoadjuvant treatment. For inter-tumor/pan-tumor analyses, gene expression was normalized to TBP (TATA-box binding protein). The TCGA pan-cancer data include thirty three (33) cancer types, and they are ACC, BLCA, BRCA,CCA, CESC, COAD, DLBC, esophageal carcinoma (ESCA), GBM, HNSC, kidney chromophobe (KICH), KIRC, KIRP, LAML, LGG, LIHC, LUAD, LUSC, MESO, OV, PAAD, PCPG, PRAD, rectum adenocarcinoma (READ), SARC, SKCM, STAD, TGCT, thyroid carcinoma (THCA), THYM, UCEC, UCS, UVM (Appendix A). In total, 11,060 samples were available for the RNA-seq data; for this study, 719 adjacent normal samples, 173 blood tumor samples of AML, 9686 primary tumors, and 395 metastatic tumors were used for analysis (Appendix A). Among the 33 cancer types, 17 cancer types had more than 5 adjacent normal samples and they were used to investigate whether there was altered gene expression in tumors compared to adjacent normal samples (Appendix A). To test the association between gene expression at mRNA and protein level, the proteomics data of BRCA [39], OV [40], and COAD [41] were downloaded from the corresponding published Appendix A.

In order to investigate the association between gene expression and drug treatment response in cancer cells, the NCI-60 database, which contains data on 60 different cancer cell lines from nine different types of tumors, was accessed using the CellMiner interface (https://discover.nci.nih.gov/cellminer/). Gene expression levels and z scores for cell sensitivity data (GI50) for 262 FDA approved drugs or drugs on clinical trials were retrieved for 59 cell lines.

### 4.2. Tumor Microenvironment Analyses

The ESTIMATE immune score and stromal score were used to analyze the infiltration levels of immune cells and stromal cells in different tumors [52]. The estimate score from this program was used to describe tumor purity. This analysis was based on the interpretation of gene expression profiles retrieved from TCGA expression data (http://bioinformatics.mdanderson.org/estimate/) [52]. Six immune subtypes were defined to measure immune infiltrate types in tumor microenvironment based on the analysis of TCGA pan-cancer data [50]. Tumor stemness features extracted from transcriptomic and epigenetic from TCGA tumor samples were used to measure stem-cell-like features of tumor cells [58].

### 4.3. Statistical Analyses

Comparisons of gene expression in all the tumors, all the adjacent paired normal, and all the metastatic tumors across all cancer types, as well as between the normal and tumors in the 17 cancer types which had more than 5 associated adjacent normal samples, were performed using linear mixed effects models. Boxplots were used to show the gene expression across cancer types and a heatmap was used to display the differential expression between tumor and normal in the tested cancer types. Univariate Cox proportional hazard regression models were used to test the association between gene expression and patient overall survival and progression free interval. Spearman or Pearson correlation was used to test the correlation between gene expression at mRNA and protein level, between gene expression and key transducers, stemness scores, stromal score, immune score, estimate score, PD1/PDL1 and CTLA-4 expression, and drug sensitivity score. Analysis of variance (ANOVA) was used to test the association between gene expression and immune infiltrate subtypes and cancer subtypes. All tests were performed using SAS9.4 (SAS Institute Inc., Cary, NC, USA). Plots were created using R (R Core Team) with packages ggplot2, pheatmap, corrplot, or survminer where appropriate [94]. We used the number of false positives method to adjust for multiple comparisons for all the tests, except for the survival study, to control the familywise error rate at α = 0.05 [53]. In more detail, we assumed 1 false positive among all the tests within each study to set the *p*-value cutoff for significance. For example, we used α = 1/(18 × 17) = 0.003 as cutoff for comparing NRP and PLXN gene expression in tumors compared to adjacent normal tissue as there were 306 tests (18 genes × 17 cancer types). For the survival study, we used α = 0.05 as cutoff without adjusting for multiple comparisons in order for the data interpretation to be consistent with the display of the results in the forest plots.

## 5. Conclusions

In the present study, we are the first to provide a systematic pan-cancer analysis of neuropilins and plexins, which are the primary receptors for the secreted proteins of class-3 semaphorin family. Overall, our study demonstrated that NRP1, NRP2, PLXNA1, PLXNA3, PLXNB3, PLXNC1, and PLXND1 were mainly associated with more aggressive phenotype of cancers and may mainly play tumor promoter roles during cancer tumorigenesis and metastasis, while the rest of the PLXN members PLXNA2, PLXNA4, PLXNB1, and PLXNB2 were more associated with better prognosis and may primarily function as tumor suppressors. Therefore, our study further confirmed their identification as potential therapeutic targets as previously reported, but in a more comprehensive point of view rather than in a limited number of cancer types. However, similar to SEMA3s, to be claimed as a specific tumor promoter or tumor suppressor for each individual member of NRPs and PLXNs, they need to be considered in a cancer type-, tumor subtype-, and immune subtype-specific manner. In summary, even though our conclusions were mainly based on the results from association studies, our work will greatly help future laboratory studies to uncover and validate their roles in tumorigenesis and drug responses, which is important for the development of personalized medicine for cancer treatment and/or to overcome drug-induced resistance.

## Figures and Tables

**Figure 1 cancers-12-01816-f001:**
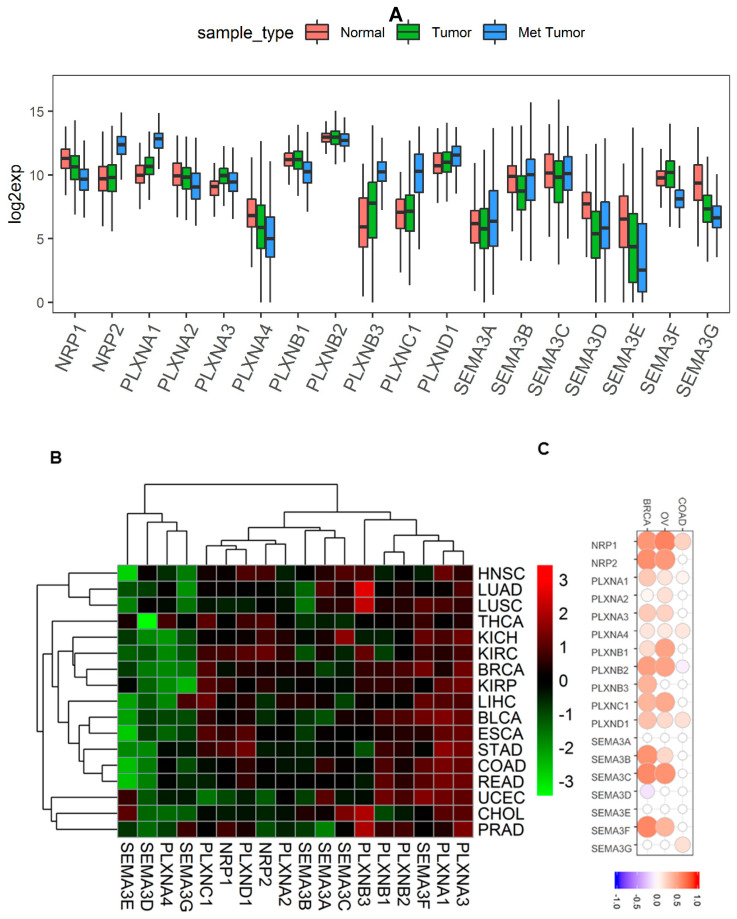
Expression levels of Class-3 semaphorins (SEMA3s) and their receptors neuropilins (NRPs) and plexins (PLXNs) in cancerous and adjacent normal tissues. (**A**). Boxplot to show the expression of SMEA3s and their receptors NRPs and PLXNs in adjacent normal (*n* = 719), primary and blood tumors (*n* = 9859), and metastatic tumors (*n* = 395) across all 33 cancer types. *p* < 1/(18 genes × 3 comparisons) = 0.018 was considered as significance. (**B**). Heatmap to show the expression difference of SEMA3s and their receptors NRPs and PLXNs comparing primary tumor to adjacent normal tissues based on log2(fold change) for 17 cancer types that have more than 5 adjacent normal samples. *p* < 1/(18 genes × 17cancer types) = 0.003 was considered as significance. (**C**). Correlation matrix to show the association between mRNA and protein level expression of SEMA3s and receptors in breast, ovarian, and colorectal cancers. *p* < 1/(18 genes × 3 cancer types) = 0.018 as significance. HNSC, head and neck squamous cell sarcoma; LUAD, lung adenocarcinoma; LUSC, lung squamous cell carcinoma; THCA, thyroid carcinoma; KICH, kidney chromophobe; KIRC, kidney clear cell carcinoma; BRCA, breast invasive carcinoma; KIRP, kidney papillary cell carcinoma; LIHC, liver hepatocellular carcinoma; BLCA, bladder urothelial carcinoma; ESCA, esophageal carcinoma; STAD, stomach adenocarcinoma; COAD, colon adenocarcinoma; READ, rectum adenocarcinoma; UCEC, uterine corpus endometrioid carcinoma; CHOL, cholangiocarcinoma; PRAD, prostate adenocarcinoma; OV, high-grade ovarian cancer.

**Figure 2 cancers-12-01816-f002:**
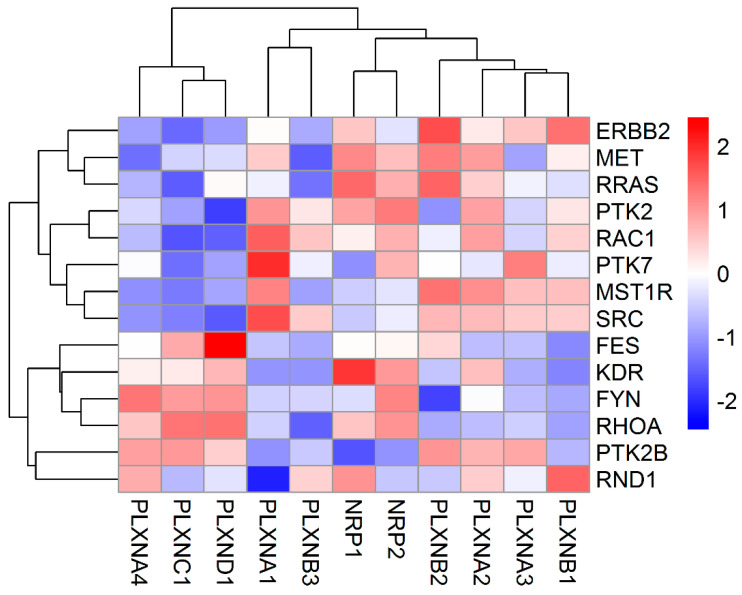
Association of NRP and PLXN family gene expression with semaphorin key signaling transducers. The heatmap was created based on the correlation coefficient between each of the SEMA3 receptors and each of the semaphorin key signal transducers averaged across all 33 cancer types from Spearman Correlation tests (*p* < 1/(11 genes × 14 key transducers) = 0.0064 as significant).

**Figure 3 cancers-12-01816-f003:**
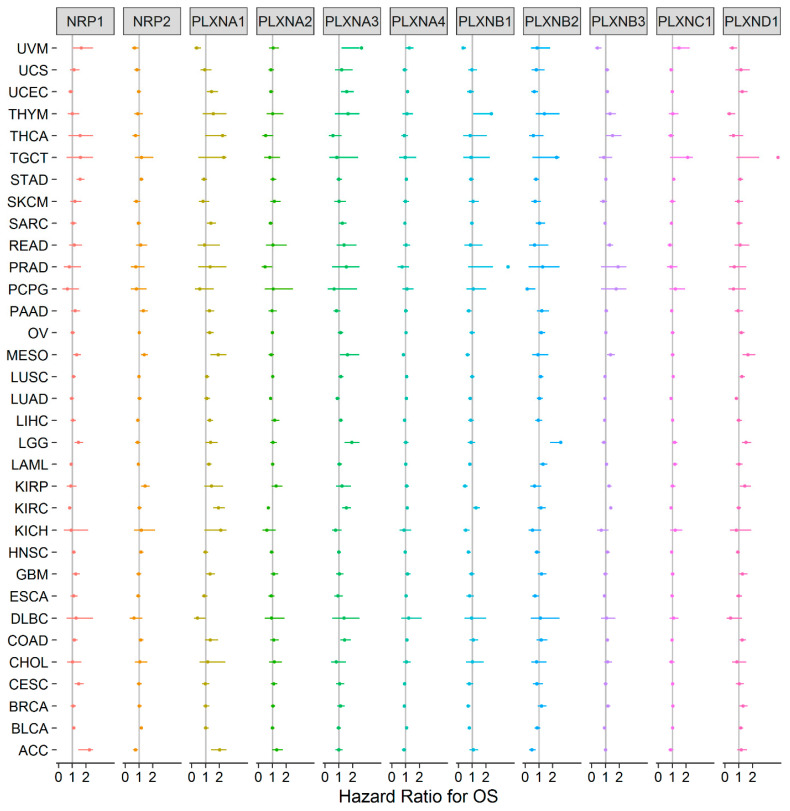
Association of NRP and PLXN family gene expression with patient overall survival for different cancer types. The forest plot with the hazard ratios (HR) and 95% confidence intervals for overall survival for different cancer types to show survival advantage (HR < 1) and disadvantage (HR > 1) with increased gene expression of NRPs and PLXNs. Univariate Cox proportional hazard regression models were used for the association tests. *p* < 0.05 was considered as significance.

**Figure 4 cancers-12-01816-f004:**
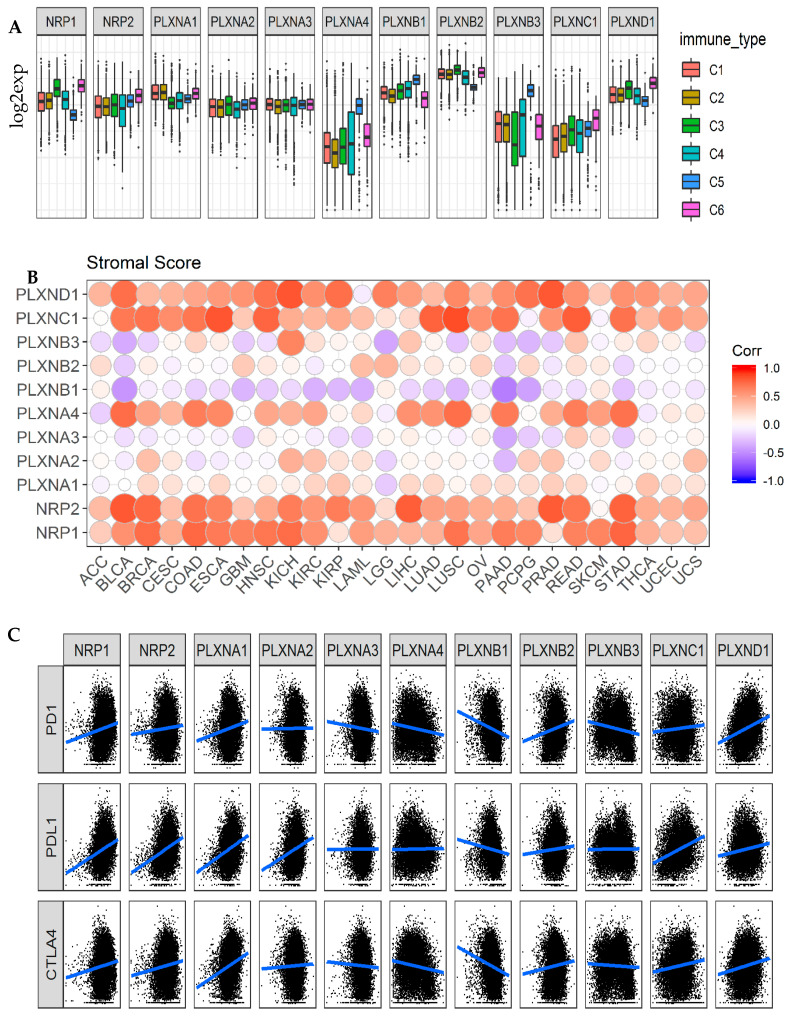
Association of NRP and PLXN family gene expression with tumor microenvironment factors. (**A**)**.** The expressions of NRPs and PLXNs within different immune infiltrate subtypes across all the cancer types tested with ANOVA. C1: wound healing, C2: INF-r dominant, C3: inflammatory, C4: lymphocyte depleted, C5: immunologically quiet, and C6: TGFβ-dominant. *p* < 1/(11 genes × 6 subtypes) = 0.015 as significance. (**B**). Correlation matrix plots to show the association between expressions of NRPs and PLXNs and stromal scores of 25 different cancer types based on ESTIMATE algorithm. Spearman correlation was used for testing. The size of the dots stands for the absolute value of the correlation coefficients. The bigger the size is, the higher the correlation is (higher absolute correlation coefficient). This also applies to Appendix A. *p* < 1/(11 genes × 26 cancer types) = 0.0034 as significance. (**C**). Scatter plots to show the correlation between the expression of NRPs and PLXNs with that of immune checkpoint molecules PD1/PDL1 and CTLA-4. Spearman correlation was used for testing. *p* < 1/(11 genes × 3 immune blockade molecules) = 0.03 as significance.

**Figure 5 cancers-12-01816-f005:**
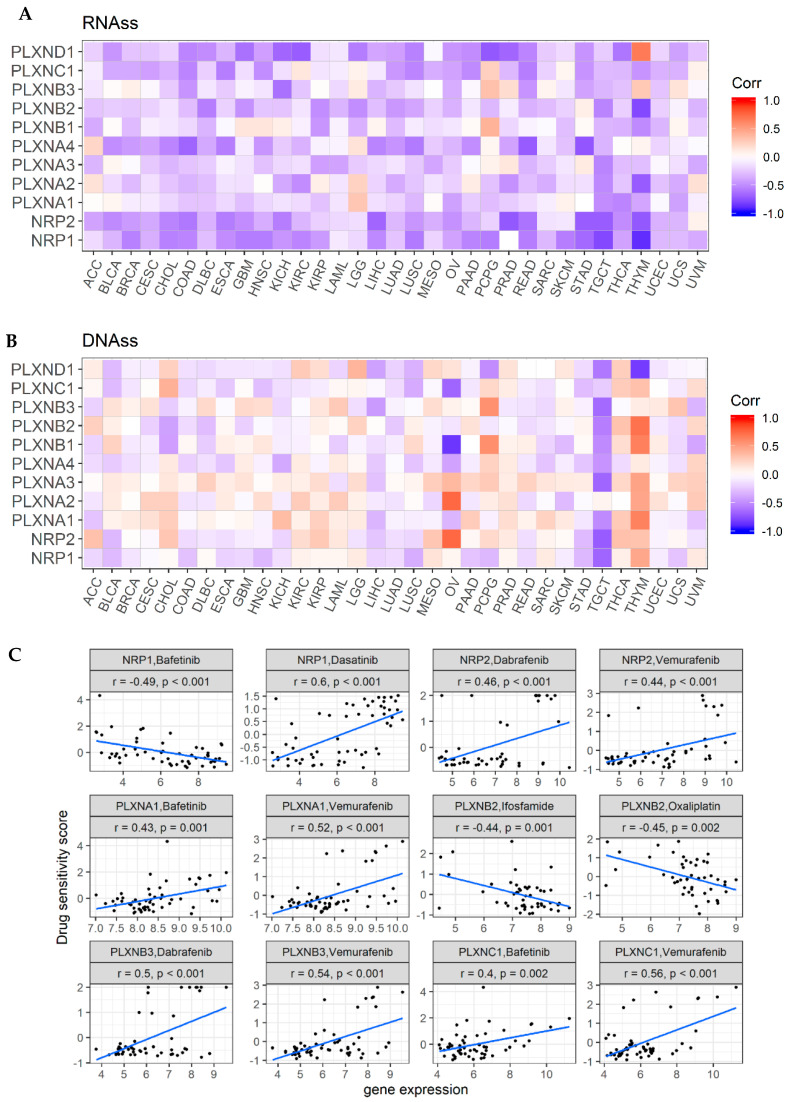
Association of expression of NRPs and PLXNs with tumor stemness and drug responses. (**A**,**B**). Correlation matrix between expressions of NRPs and PLXNs and cancer stemness scores RNAss (**A**) and DNAss (**B**), respectively, based on Spearman correlation tests. (**C**). Scatter plots to show the association between expression of NRPs and PLXNs and drug sensitivity (*Z*-score from CellMiner interface) tested with Pearson Correlation using NCI-60 cell line data. *p* < 1/(11 genes × 26 cancer types) = 0.003 for A and B, and *p* < 1/200 (drugs) = 0.005 for C was considered as significance.

**Figure 6 cancers-12-01816-f006:**
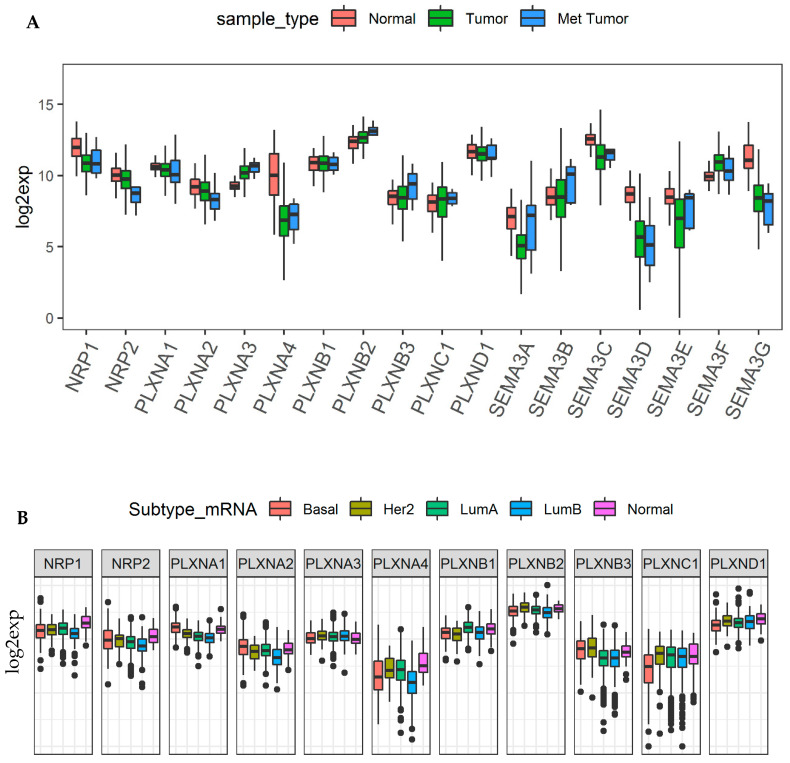
NRPs and PLXNs in breast cancer tumors and different molecular subtypes. (**A**). Box plots to show the expression of SEMA3s and their receptos NRPs and PLXNs in primary tumors (*n* = 1095), metastatic tumors (*n* = 7), and adjacent normal tissue samples (*n* = 113) of the TCGA breast cancer cohort (BRCA). Linear mixed effects models were used for testing. (**B**). Association of NRP and PLXN gene expression with breast cancer molecular subtypes (*p* < 0.0001) tested with ANOVA. *p* < 1/(18 genes × 3 comparisons) = 0.018 and <1/(11 genes × 5 subtypes) = 0.015 was considered as significance for A and B, respectively.

**Figure 7 cancers-12-01816-f007:**
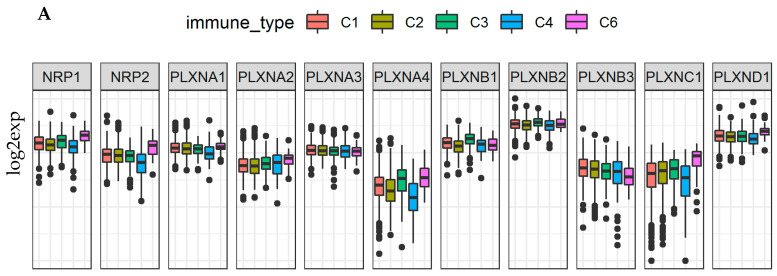
NRPs and PLXNs are associated with immune subtypes, tumor infiltrates, and patient survival in breast cancer (BRCA). (**A**). Association of NRP and PLXN gene expression with immune infiltrate subtypes in breast cancer tested with ANOVA (*p* < 0.0001). C1: wound healing, C2: INF-r dominant, C3: inflammatory, C4: lymphocyte depleted, C5: immunologically quiet, and C6: TGFβ-dominant. *p* < 1/(11 genes × 5 subtypes) = 0.018 as significance. (**B**). Correlation matrixes between NRP and PLXN gene expression and RNAss, DNAss, stromal score, immune score, and estimate score. Spearman correlation tests were used for testing. *p* < 1/(11 genes × 5 scores) = 0.018 as significance. (**C**). The forest plots with the hazard ratios (HR) and 95% confidence intervals for overall survival (OS) and progression free interval (PFI) to show survival advantage (HR < 1) and disadvantage (HR > 1) with increased gene expression of NRPs and PLXNs in BRCA. Univariate Cox proportional hazard regression models were used for the association tests. *p* < 0.05 was considered as significance.

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
