# Peer review of "Characterization of Class-3 Semaphorin Receptors, Neuropilins and Plexins, as Therapeutic Targets in a Pan-Cancer Study"

_cancers, 2020, doi:10.3390/cancers12071816_

Round 1

Reviewer 1 Report

The authors have adequately addressed to the concerns raised previously by this reviewer.

This manuscript is a resubmission of an earlier submission. The following is a list of the peer review reports and author responses from that submission.

Round 1

Reviewer 1 Report

This article describes the expression of the receptors of SEMA3 family and their expression levels in a pan-cancer analysis and follows previous publications describing similar studies in SEMA3 family members. The authors examine SEMA3 receptor expression and correlate overall survival and progression free interval of patients in addition to exploring expression of these molecules with tumor microenvironment factors, specifically immune signatures. Finally, the authors perform similar analysis on breast cancer samples and further break down SEMA3 family receptor expression by sub-type of breast cancer. The overall findings conclude that SEMA3 family receptor expression is highly heterogenous, varies across cancer types, and suggest that receptors can act in either a tumor promotional or tumor suppressive manner in a cancer specific, and often subtype specific, manner.

Overall comments:

  • While most of the text is well written, the article could benefit from being shortened.
  • Line 45 correct SMEA3s to SEMA3s
  • Conflicting citations 13+37 are these duplicate references?
  • Indicate statistical significance within each figure panel (main and supplement) so the reader can more easily digest the findings and conclusions stated in text

Figure 1

  • ‘A’ label is missing
  • If possible present data in panel A in the order normal, tumor, then metastasis
  • The current labeling for the cancer types in the heatmap in panel B is only defined in the supplement, please move to figure ledged or add less technical abbreviations.

Figure 2

  • Increase bar/line size for hazard ratios
  • If possible, have even distribution of x-axis across all receptors (ie so 1 is in the middle)

Figure 3

  • Panel A y axis label is missing
  • Either in panel A or in the figure ledged define immune types
  • For panel C please include a table of the statistics for correlation analysis (supplemental)
  • Re-word section describing panel C giving more detail/justification why these correlation analyses were performed

Figure 4

  • No comments

Figure 5

  • If possible present data in panel A in the order normal, tumor, then metastasis (as in Fig 1)
  • Panel B y axis label is missing

Figure 6

  • As in Fig 3 please define immune sub-types
  • Panel A y axis label is missing
  • As in Fig 3 please add a supplementary table with statistics for panel B
  • Remove BRCA from x-axis label in panel C, can be easily confused with BRCA gene status in breast cancer and it is clear that this entire figure is breast cancer only.

Table S1

  • Fix text wrap so words are not broken-off

Figure S1

  • Alter box plots size so color is visible in across all cancer types and receptors

Figure S3

  • May be a technical issue, but all labels are missing from this figure on the copy I received.

Table S2

  • Add drug targets/MOA to table

Pending the recommended changes, I approve this manuscript for publication.

Reviewer 2 Report

The knowledge about the relationship between Semaphorin and cancer is new.

There is lack of data and understanding of its mechanism of action

Semaphorin is highly potential molecule for diagnosis and targeted treatment.

There is high volume of study and results in this manuscript regarding the above mentioned relationship.

Reviewer 3 Report

This is a sounding and interesting manuscript which unfortunately suffers from the recent publication by the same group. Indeed, the used bioinformatic tools have already been developed and herein the authors have only extended to a larger cancer context with no evidence that really strikes from the already published observations. 

Despite the bioinformatic analyses are well and properly performed the advancement in the knowledge compared to the previous publication is minimal and consequently does not warrant publication in Cancers.

The authors need to perform an extensive amount of work to make the manuscript more novel,  for instance complementing their analysis on gene expression with protein expression and  evaluation of the key transducers of the activated signalling in human cancers. This would surely increase the impact of the manuscript and make it a strong candidate for being published in Cancers.